# Anthropometric and Functional Profile of Selected vs. Non-Selected 13-to-17-Year-Old Soccer Players

**DOI:** 10.3390/sports8080111

**Published:** 2020-08-09

**Authors:** Erik Nughes, Vincenzo Rago, Rodrigo Aquino, Georgios Ermidis, Morten B. Randers, Luca Paolo Ardigò

**Affiliations:** 1Ministry of Education, Universities and Research, 00153 Rome, Italy; eriknughes@hotmail.it; 2Portugal Football School, Portuguese Football Federation, 1495-433 Oeiras, Portugal; vincenzo.rago@fpf.pt; 3Faculty of Health Sciences and Sports, Universidade Europeia, 1500-210 Lisbon, Portugal; 4Research Group in Soccer Science, Center of Physical Education and Sports, Department of Sports, Federal University of Espírito Santo, Vitória 29075-910, Brazil; aquino.rlq@gmail.com; 5Department of Movement Sciences and Wellness, University of Parthenope, 80133 Naples, Italy; germidis1990@gmail.com; 6Department of Sports Science and Clinical Biomechanics, Faculty of Health, University of Southern Denmark, 5220 Odense, Denmark; mranders@health.sdu.dk; 7School of Sport Sciences, The Artic University of Norway, 9019 Tromsø, Norway; 8School of Exercise and Sport Science, Department of Neurosciences, Biomedicine and Movement Sciences, University of Verona, 37131 Verona, Italy

**Keywords:** performance, dribbling, height, sprint, repeated sprint ability, agility

## Abstract

The purpose of this study was to compare anthropometric and functional profiles of 13-to-17-year-old soccer players according to their competitive level. Height, body mass, percentage of body fat, countermovement jump height, change of direction ability, 5- and 15-m sprint times, repeated sprint ability (RSA), intermittent recovery performance, and dribbling skills were collected in 115 young Italian soccer players. Players were divided into selected (i.e., competing at national level, n = 17 U15 and 47 U17) and non-selected (i.e., competing at regional level, n = 43 U15 and 8 U17) groups. U17 selected players were taller, quicker over 5 and 15 m, more agile, and had better RSA, prolonged intermittent recovery ability, and dribbling skills than their non-selected counterparts (*d* = 0.28–0.55, *p* < 0.05). In particular, selected players showed lower times on the first three and the last shuttle of the RSA test (*d* = 0.28–0.34, *p* < 0.05). No significant differences were observed in U15 players (*p* > 0.05). Discriminant analysis revealed that dribbling skills, 15-m sprint time, and height best discriminate U17 players by competitive level (*p* < 0.001). Anthropometric characteristics and functional abilities can discriminate across competitive standards between male U17 but not U15 soccer players. In particular, these findings suggest the importance of dribbling skills, 15-m sprint, and height in U17 players.

## 1. Introduction

Soccer performance is complex and multifactorial, depending on a number of different variables such as anthropometric profile, functional capacity, psychological factors, and technical-tactical aspects, among others [1]. Soccer scouts often observe and select young players based on match performance [2], which can be affected by opponent rank, current match status, and other situational factors [3]. The use of standardized measurements (e.g., anthropometric, fitness, and technical skills) can assist coaches and practitioners in objectively examining soccer players’ characteristics for subsequent selection [4,5,6].

Extensive research has shown that anthropometric characteristics and performance in field tests effectively discriminate soccer players by competitive level [7,8,9]. In general, selected (i.e., professional) young players are taller, heavier, and more biologically mature than their non-selected (i.e., amateur) counterparts [8,10,11]. Notwithstanding, when it comes to performance in field tests, selected players are commonly more powerful (e.g., as from jump and sprint tests; [8,12,13]) and agile (e.g., capable to quickly change direction within confined spaces) and have better cardiorespiratory fitness (as from either field and laboratory time-to-exhaustion tests) and technical skills than their non-selected counterparts [8]. Muscle power and cardiorespiratory fitness, which are associated with physical performance (e.g., amount of high-speed distance, >18 km·h^−1^ [14]) during competitive youth soccer games [3], were considered relevant. Beyond these, limited attention has been paid to repeated sprint ability (RSA), which has been defined as the ability to recover and reproduce performance over subsequent sprints.

The importance of RSA is underlined by the fact that the occurrence of repeated sprints is highly taxed during youth soccer, with meaningful activity declines observed toward end of match play [15]. Additionally, studies have demonstrated good ecological validity of RSA in relation to physical performance during matches [16] and good sensitivity in discriminating between competitive standard [16,17] and age-group. However, research describing fitness profiles of young soccer players has predominantly focused on power-based abilities (e.g., jump, sprint, and change of direction (COD) ability) and cardiorespiratory fitness (e.g., Yo-Yo Intermittent Recovery Test) but with limited attention paid to RSA [18,19,20,21]. Whereas these studies have considered RSA mean, best time, and rate of performance decrement throughout the test as indicators of RSA, only one study has provided information on every single shuttle [16]. This could be of interest, as professional players seem to pace their effort to manage fatigue during soccer training [22]. Additionally, RSA might be considered a physiologically complex phenomenon to understand. Indeed, repeated sprint efforts require a combination of both anaerobic capacity, required to maximally sprint during each shuttle, and aerobic capacity, to efficiently recover between sprints maintaining low metabolic strain (e.g., oxygen consumption between sprints). However, information during standardized exercise (e.g., RSA testing) is unknown.

Given the above, further understanding of determinants of success, such as specific anthropometric, functional capacity, and technical skills of players, may be important in both talent identification and development processes. Thus, the aim of the present study was to compare anthropometric and functional profiles of young soccer players according to their competitive level, with special emphasis on the identification of variables that best differentiate by competitive level.

## 2. Materials and Methods

All players were evaluated at the start of the competitive period (September 2017) for their respective clubs. Protocol included interview, anthropometry, field-based physical tests, and dribbling skill test. Each player was tested on two occasions within a one-week period. Anthropometric dimensions, dribbling skills, sprint time, COD, and RSA were assessed during the first visit in listed order. Countermovement jump (CMJ) and cardiorespiratory fitness were assessed during second visit. With exception of anthropometry, all tests were administered outdoor on a 3rd generation artificial grass field. Prior to testing physical and dribbling skills, players performed 12-min warm-up consisting of jogging and stretching exercises. Players were familiar with these assessments from previous season and wore soccer boots during all field tests.

One hundred and fifteen young soccer players (60 U15 and 55 U17 players) from three local different soccer clubs (Campania, Italy; one elite, two non-elite) were assessed, as already requested by their coaches. Players were divided into selected group (*n* = 17 U15 and 47 U17), competing at national level and non-selected group (*n* = 43 U15 and 8 U17) competing at regional level. Study was approved by Institutional Ethics Board of Medical School of Ribeirão Preto of University of São Paulo (protocol number 711.038). Players and their parents or legal guardians were informed about benefits and possible risks associated with study

### 2.1. Anthropometry

The same experienced author measured body mass, height, sitting height, and four skinfolds (triceps, suprailiac, abdominal, and thigh). Body mass was measured to nearest 0.1 kg with a scale (model 770, SECA, Hanover, MD, USA). Height was measured to nearest 0.1 cm with a Harpenden stadiometer (model 98.603, Holtain Ltd., Crosswell, UK). Body mass index was calculated from height and body mass. Skinfolds were measured to nearest mm with a Lange caliper (Beta Technology, Ann Arbor, MI, USA). Percentage of body fat was estimated using sum of skinfolds (*r* = 0.77–0.88 with hydrostatic weighting [23]). Each measurement was taken three times and median was used for further analysis. Players wore light clothing and shoes were removed.

### 2.2. Countermovement Jump

Jump height was collected using an accelerometric system (Myotest SA, Sion, Switzerland) sampling at 500 Hz. Device (dimensions 5.4 × 10.2 × 11.1 cm, weight 58 g) contains a 3D inertial accelerometer (full scale 68 g) that allows vertical acceleration to be recorded as previously described [24]. Device was perpendicularly attached to a large (8.5 cm) Velcro elastic belt and fixed at hip level on left side of body, as suggested by manufacturer. Participants performed three CMJ trials separated by ~2 min rest, ensuring hands were kept in contact with hips throughout the jump. As suggested by Domire et al. [25], participants executed jump by bending knees to a position that they found to be comfortable (i.e., preferred starting push-off position). Best trial (higher jump height) was used for further analysis. This variable has shown good test–retest reliability (intra-class correlation coefficient (ICC) = 0.97, coefficient of variation (CV) = 4.2 [24]).

### 2.3. Sprint

Split sprint times were measured at 5 and 15 m during a 20-m sprint test. Elapsed times were measured using three pairs of photoelectric cells (Witty, Microgate, Bolzano, Italy) placed at starting line and at 5 and 15 m [8]. Players were instructed to run as fast as possible from standing position 30 cm behind starting line up to a cone placed 20 m ahead. This strategy was based on practical experience since players tend to slow down before reaching finish. The fastest sprints were used for further analysis. Trials were interspersed by >3 min of passive rest.

### 2.4. Change of Direction

The Arrowhead test was used to evaluate COD ability. A trial repetition was administered at moderate pace once before starting actual test. Players were instructed to run as fast as possible from a standing position 30 cm behind starting line. Used photocells (Witty, Microgate, Bolzano, Italy) were placed next to start/finish line (start and finish shared same line). Test was carried out along a path that recalls the shape of an arrowhead, from which it takes its name. Participants sprinted and changed direction maximally throughout test, turning around and not above a cone. Otherwise, trial was stopped and reattempted. Two trials in total were completed (counterclockwise and clockwise COD). Order was randomized among participants to avoid any effect of fatigue on any side. Participants started when ready, thus eliminating reaction time, and completed two trials interspersed by at least 3 min of rest. Best of two trials for each side was used for further analysis. Reliability of test was verified in a recent study (ICC = 0.80–0.83, CV = 1.25%–2.21%, [26]).

### 2.5. Repeated Sprint Ability

A soccer-specific RSA test consisting of six 40-m (20 + 20 m) shuttle sprints interspersed by 20 s of passive recovery was adopted [16]. Athletes started from a line, sprinted for 20 m, touched a line with a foot, and came back to starting line as fast as possible. After 20 s of passive recovery, soccer player started again. Each single shuttle time was recorded using a photocells system (Witty, Microgate, Bolzano, Italia). Five seconds before start of each sprint, subjects assumed “ready” position and waited for start signal. Players were instructed to provide maximal effort over each shuttle and not adopt any pacing strategies. RSA mean time, best time, and decrement percentage over sprints were calculated (CV = 0.8–1.2 [27]).

### 2.6. Yo-Yo Intermittent Recovery Test Level 1

Ability to perform prolonged intermittent exercise was assessed by means of Yo-Yo Intermittent Recovery Test Level 1 (YYIR1), which consists of repeated 2 × 20-m runs (shuttles) interspersed by 10-s rest periods and speed progressively increasing, controlled by audio bleeps from a tape recorder [28]. Goal of test was to perform as many shuttles as possible. When player failed twice to reach finish line in time, test stopped and last fully completed shuttle was considered final one. Test was performed only once. Total distance covered over YYIR1 is considered test score and outcome of ability to perform intermittent exercise (correlation with laboratory-measured maximal oxygen uptake = 0.70, test–retest correlation = 0.93, CV = 8.1% [28]).

### 2.7. Technical Skills

Short dribbling test was used to evaluate technical skills related to coordination and speed with ball at foot [29]. Player starts with one foot on starting line and the other foot behind it with the ball (Figure 1). Player follows a planned path consisting in dribbling a set of cones as fast as possible. A trial repetition was administered at moderate speed before starting actual test (made of two repetitions at maximum speed interspersed by ~2 min rest). The trial was not valid if player let a cone fall or followed wrong direction. The best repetition (i.e., shorter time) was used for further analysis.

### 2.8. Statistical Analysis

Shapiro–Wilk test revealed that data within competitive levels were normally distributed (*p* > 0.05). Differences between selected and non-selected players were assessed using independent sample *t* tests. The *t* statistic was then completed with calculation of effect size (*d*) and associated 95% confidence intervals (CIs) to qualitatively interpret magnitudes of differences [30]. After identifying variables that were statistically significant between competitive levels, forward stepwise discriminant function analysis was used to identify smallest variables set that maximized differences between groups. Finally, results from classification matrix revealed how accurate smallest variables set obtained from discriminant function was in recovering original grouping of all subjects.

Data were reported as mean ± standard deviation (SD) for each variable. Significance was set at *p* < 0.05. Data analysis was performed using Statistical Package for Social Science software (version 23, IBM SPSS Statistics, Chicago, IL, USA) and a customized Excel spreadsheet.

## 3. Results

### 3.1. Anthropometric Characteristics

U17 selected players were moderately taller than non-selected players (*d* = 0.35 (0.08; 0.56); *p* = 0.009). On the other hand, U15 players showed similar heights between competitive levels (*p* = 0.536). No significant differences between competitive levels were observed in body mass, BMI, and fat percentage (*p* > 0.05). Detailed description of anthropometric characteristics of 13-to-17-year-old soccer players is reported in Table 1.

### 3.2. Functional Capacity

U17 selected players were more agile, quicker over 5 and 15 m, more able to perform prolonged intermittent exercise, and skillful in the dribbling test than their non-selected counterparts (*d* = 0.28 to 0.55, *p* < 0.05). Regarding RSA, selected U17 players had a lower mean and best time throughout the test (*d* = 0.28 (0.01; 0.51) and 0.34 (0.07; 0.56), respectively; *p* < 0.05) but not in decrement percentage (*p* > 0.05). Specifically, selected players showed lower times on the first, second, third, and sixth shuttles (*d* = 0.28 to 0.34, *p* < 0.05, Figure 2). No significant differences in functional capacity were observed between selected and non-selected U15 players. A detailed description of functional capacity of 13-to-17-year-old soccer players is reported in Table 2.

### 3.3. Discriminant Function Analysis

The importance of variables which most discriminate by competitive level is ranked as follows: dribbling skills, 15-m sprint time, and height (*p* < 0.001). A detailed description of discriminant function analysis is reported in Table 3. From the classification matrix results, 80% of selected and 69% of non-selected players were correctly classified into their original group, respectively. The percentage of correct classifications was 75%.

## 4. Discussion

The main findings of this study were that U17 selected soccer players were taller and presented, in general (except for CMJ), better functional capacities than their non-selected counterparts. In addition, combined effects of dribbling skills, 15-m sprint time, and height accounted for 75% of correct classifications of selected U17 players. In the present study, selected players were moderately taller compared to their non-selected counterparts. This is in line with previous research showing that adult players attaining higher levels of play were, on average, substantially differentiated from amateur players in height as well as body mass [31]. However, beyond height, other anthropometric measures did not show any significant differences between competitive levels. Previous systematic review showed weak correlations between anthropometric measures and match running demands in young players [3]. Therefore, in practical settings, anthropometric variables may well guide training programs, but these tests do not seem to have the best potential to associate with competitive level and match performance.

High-intensity activities are fundamental aspects for performance in soccer [32]. The analysis of power-based tests in our study revealed that, irrespectively of age group, there were no significant differences in CMJ between selected and non-selected players. On the other hand, U17 selected players were capable of higher acceleration over both 5 m and 15 m than non-selected players. Accordingly, previous studies showed that elite outfield players tended to present better COD and linear sprint (15-, 25-, and 30-m) performances than non-elite ones [8,26]. Differences in sprint time over 5 and 15 m might be also relevant in relation to the fact that elite players predominantly perform their high-speed runs over short distances during the match [33]. In particular, sprint time over 15 m was the second variable that best discriminated competitive levels (see Table 3). Therefore, coaches and practitioners should consider this ability over the talent identification and development process. Better RSA results observed in selected compared with non-selected U17 players confirmed previous findings regarding this matter showing better RSA performance in elite young players from different age groups compared with amateur players [7,17,27]. In elite young soccer players, significant correlations have been reported between distance covered during official matches and mean sprint times in RSA tests [16]. Our within-test detailed analysis revealed that U17 selected players show better times over the first three and the last shuttles. This may possibly indicate that, despite players being required to provide maximal effort in all shuttles, U17 selected players might be able to better recover during resting periods between shuttles. This could be clarified by physiological measurements over RSA test shuttles to feature fatigue-induced performance decay throughout protocol.

During matches, more than 90% of the time is spent making use of aerobic metabolism and the average intensity is around 80%–90% of maximal heart rate [34]. Therefore, the aerobic power of players is a factor which should be highly considered by coaches and practitioners over the talent identification and development process. In the current study, U17 selected players presented longer distances covered during YYIR1 compared to non-selected ones. However, this variable did not factor into the forward stepwise discriminant function analysis supporting coaches and scouting departments’ attitudes to not predominantly consider ability to perform prolonged exercise. Nevertheless, previous research showed that peak maximal oxygen uptake is higher in elite league teams’ players compared with the first division. This is probably due to the fact that the level of competition in the elite league is higher. Its players are more likely to be exposed to a higher level of competition and this leads them to a more advanced development of aerobic capacities [35]. In addition, Mohr et al. [36] showed that top level players covered 10% more distance in YYIR1 than medium level. In light of this study’s results, aerobic endurance is anyway an important physical attribute of competitive level in soccer. In the present study, technical skill was the main distinguishing factor between selected and non-selected players. This is supported by previous research indicating that better players distinguish themselves by their running speed while dribbling the ball [37,38,39]. Indeed, frequent involvement with the ball and successful performance in skill-related activities are important determinants of success in soccer [9,39].

The main limitation of this study is the lack of an assessment of biological maturation. This study investigated 13–15 and 15–17-year-old players who were homogeneous in terms of chronological age. Nevertheless, they were still in the process of growth and maturation and this could have interfered with their physical test measures. Moreover, players were evaluated at the start of the competitive period (September), when it is likely to have a relative age effect bias with players born earlier in their birth year, who might present physical advantages compared with players born later in the same year [40]. Furthermore, we did not consider playing position. Soccer players must adapt to match the physical demands featuring their specific positional roles. Indeed, professional senior fullbacks, wide midfielders, and forwards covered total distances over match-play longer than other positional groups [16,41]. In particular, the occurrence and nature of repeated sprints during youth match-play are affected by age, field position, and playing time, with wide players displaying higher repeated-sprint activity [15]. Nevertheless, to the best of the authors’ knowledge, no previous studies have investigated the interaction effects of competitive level and playing position in young soccer players using an array of tests to help assess multivariate profiles (e.g., anthropometric measures, physical performance, and technical skills). Finally, the lack of differences in the CMJ test could be due to the fact that CMJ performance is strictly dependent on jump execution. In other words, we did not consider various temporal and force components throughout various phases preceding flight, which might better explain the neuromuscular characteristics of athletes [42].

The lack of differences in the U15 group could be due to the fact that U15 players were within an age interval when individual differences are at their peak, which might skew the prediction of success in adolescent soccer players [31]. Indeed, evidence suggests that >70% of professional soccer player started their careers at elite level between 17 and 20 years of age [43]. Additionally, differential experience and training time between elite and non-elite players might have contributed to the observed differences in functional capacities. Summarizing, this article provides new insights into selection criteria adopted by coaches and scouting of young soccer players. Specifically, dribbling skills, 15-m sprint time, and height appear to be the most adopted selection criteria. On the other hand, adopted measurements appear to be not indicative of U15 selection.

## 5. Conclusions

Anthropometric characteristics and functional capacities can discriminate across competitive standards between male U17 but not U15 soccer players. Compared with non-selected U17 soccer players, selected ones appeared to be moderately taller, more agile, quicker over short distances (e.g., 5 to 15 m), more able to repeat multiple sprints, perform prolonged intermittent exercise, and efficiently dribble. In particular, dribbling skills, 15-m sprint time, and height correctly classified 75% of selected U17 players. This suggests the usefulness of these measures in distinguishing competitive level in this age group. Therefore, coaches and practitioners could consider these abilities over the talent identification and development process.

## Figures and Tables

**Figure 1 sports-08-00111-f001:**
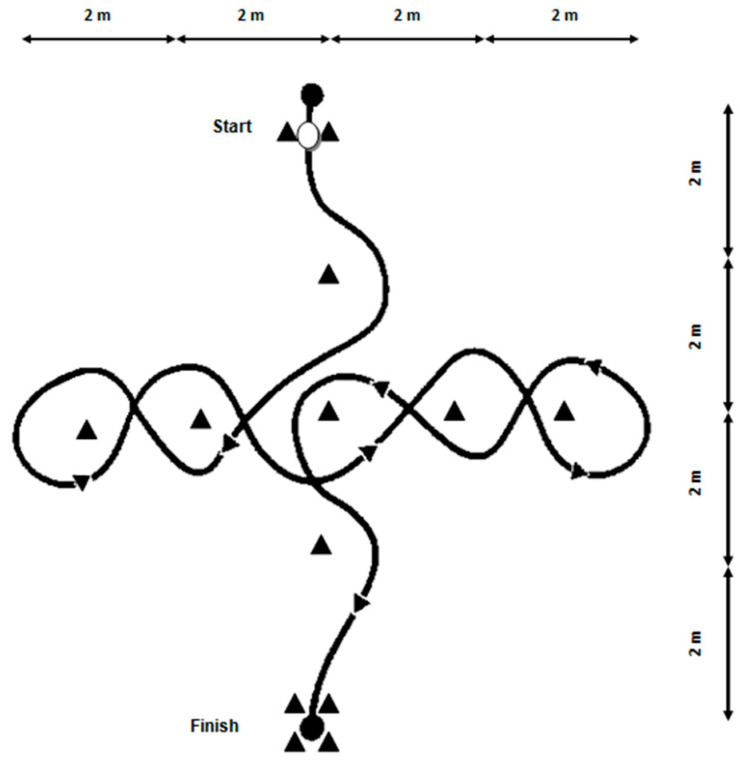
Short dribbling test layout.

**Figure 2 sports-08-00111-f002:**
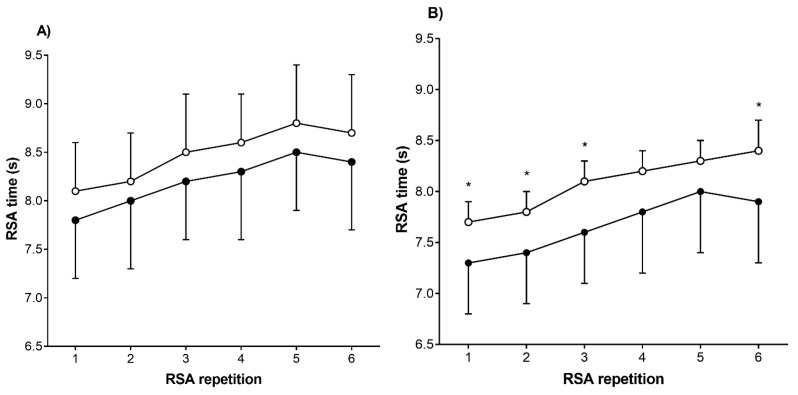
Detailed descriptive analysis of repeated sprint time between competitive level in (**A**) U15 and (**B**) U17 soccer players. Blank circles are non-selected players. Black circles are selected players. RSA repeated sprint ability, * significantly different than selected players (*p* < 0.05).

**Table 1 sports-08-00111-t001:** Anthropometric characteristics (mean ± SD) of 13-to-17-year-old soccer players.

Variable	Age Group	Non-Selected	Selected	*d* (95% CIs)	*p*
Body mass (kg)	U15	59.2 ± 9.8	55.6 ± 6.9	0.17 (−0.08; 0.41)	0.175
	U17	63.9 ± 9.7	65.1 ± 7.8	0.05 (−0.21; 0.32)	0.682
Height (cm)	U15	166.0 ± 13.5	168.2 ± 7.4	0.08 (−0.18; 0.33)	0.536
	U17	168.6 ± 4.5	174.7 ± 6.0	0.35 (0.08; 0.56)	0.009
BMI (kg·m^−2^)	U15	22.6 ± 10.5	19.6 ± 1.7	0.15 (−0.11; 0.39)	0.252
	U17	22.4 ± 3.1	21.2 ± 2.0	0.19 (−0.07; 0.44)	0.153
Body fat (%)	U15	8.6 ± 5.1	6.8 ± 3.1	0.16 (−0.09; 0.40)	0.197
	U17	8.7 ± 4.3	7.4 ± 3.1	0.14 (−0.12; 0.40)	0.287

BMI body mass index, CI confidence interval.

**Table 2 sports-08-00111-t002:** Functional capacity (mean ± SD) of 13-to-17-year-old soccer players.

Variable	Age Group	Non-Selected	Selected	*d* (95% CIs)	*p*
CMJ (cm)	U15	35.2 ± 5.4	34.1 ± 3.6	0.11 (−0.16; 0.35)	0.420
	U17	37.4 ± 3.4	36.9 ± 4.6	0.04 (−0.23; 0.31)	0.768
AAT(s)	U15	18.66 ± 0.95	19.07 ± 0.67	0.21 (−0.05; 0.45)	0.104
	U17	18.05 ± 0.77	17.08 ± 1.15	0.30 (0.03; 0.53)	0.026
5-m sprint time (s)	U15	1.23 ± 0.09	1.22 ± 0.08	0.07 (−0.19; 0.32)	0.592
	U17	1.21 ± 0.08	1.13 ± 0.08	0.36 (0.10; 0.57)	0.007
15-m sprint time (s)	U15	2.85 ± 0.18	2.80 ± 0.13	0.12 (−0.14; 0.37)	0.368
	U17	2.71 ± 0.12	2.56 ± 0.14	0.38 (0.12; 0.59)	0.005
RSA mean time (s)	U15	8.42 ± 0.51	8.15 ± 0.63	0.22 (−0.04; 0.45)	0.090
	U17	8.02 ± 0.20	7.62 ± 0.53	0.28 (0.01; 0.51)	0.037
RSA best time (s)	U15	8.03 ± 0.45	7.81 ± 0.63	0.19 (−0.07; 0.43)	0.142
	U17	7.71 ± 0.20	7.26 ± 0.47	0.34 (0.07; 0.56)	0.012
RSA decrement (%)	U15	5.7 ± 2.4	5.1 ± 1.3	0.13 (−0.13; 0.38)	0.321
	U17	5.1 ± 1.1	5.5 ± 2.3	0.06 (−0.21; 0.33)	0.643
YYIR1 (m)	U15	763 ± 236	810 ± 287	0.09 (−0.18; 0.34)	0.517
	U17	950 ± 402	1303 ± 435	0.28 (0.01; 0.51)	0.037
Dribbling skills (s)	U15	14.39 ± 2.12	13.50 ± 3.50	0.16 (−0.10; 0.40)	0.229
	U17	14.26 ± 0.64	12.74 ± 0.84	0.55 (0.33; 0.72)	<0.001

CMJ countermovement jump, AAT arrowhead agility test, RSA repeated sprint ability, YYIR1 Yo-Yo Intermittent Recovery Test Level 1, CI confidence interval.

**Table 3 sports-08-00111-t003:** Summary of stepping summary in forward stepwise discriminant analysis in U17 soccer players.

Step		Wilks’ Lambda	Approx. *F*-Ratio	*p*
1	Dribbling skills	0.693	23.471	<0.001
2	15-m sprint time	0.618	16.04	<0.001
3	Height	0.562	13.228	<0.001

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
