# Peer review of "Anthropometric and Functional Profile of Selected vs. Non-Selected 13-to-17-Year-Old Soccer Players"

_sports, 2020, doi:10.3390/sports8080111_

Round 1

Reviewer 1 Report

Lines 40-44. These sentences should be merged for improved reading flow.  Suggested rewrite. “Soccer scouts often observe and select young players based on match performance [2], which can be affected by opponent rank, current match status, and other situational factors [3]. The use of standardized measurements (e.g., anthropometric, fitness, and technical skills) can assists coaches and practitioners in objectively examining soccer players' characteristics for subsequent selection [4-6].”

Lines 52-55. Very confusing sentence in its current structure. Rewrite and separate into two separate sentences so the point(s) being made are clear. “Beyond the importance of muscle power and cardiorespiratory fitness, which are associated with physical performance (e.g., amount of high-speed distance) during competitive youth soccer games [3], limited attention has been paid to repeated sprint ability (RSA), which has been defined as the ability to recover and reproduce performance over subsequent sprints.” Additionally, does “amount of high-speed distance” refer to “total game running volume at     ?       velocity”?  

Line 59. Add play after the word “match”

Line 62. Add an example for CR fitness, (YO-YO intermittent test?)

Line 77.  Edit to read this for improved reading flow. “…evaluated at the start of the competitive (September 2017) for their…”

Line 80. Edit by adding  the “…during the first…”

Line 94. How is the person “trained”? Suggest replacing with “experienced” unless the trained is going to be explained such as trained by a specific organization.

Line 114. Edit sentence. “Split sprint times were measured at 5 and 15-m during a 20-m sprint test.”

Line 118. Remove “Better” and just state, “The fastest sprints…”

Dribbling test would benefit from a figure that displays the planned path.

Line 157. There should be a subsection title of Statistical Analysis

Line 202.  What does higher mean? Confusing sentence structure as it is difficult to figure out the point. Probably could be easily rewritten for clarity.

Line 205. Change of terms from “Selected” to “Elite”, be consistent with terms.

Line 217. The 5-m and 15-m times represent acceleration not agility or quickness unless the athlete’s movement time to a stimulus was recorded.

Lines 239-242. The sentence is confusing and needs to be rewritten for clarity and flow. “Nevertheless, previous research showed peak maximal oxygen uptake in elite league teams’ players higher than in first division probably due to elite league’s higher level of competition, which players were more likely exposed to, leading them thus to a more advanced development of aerobic capacities [34].”

Line 250-253. Rewrite these sentences for improved flow. “Main limitation of this study is lack of biological maturation assessment. In spite of being homogeneous in terms of chronological age, investigated 13-15 and 15-17 yrs old players were still in the process of growth and maturation and that could have interfered with their physical test measures.”

Author Response

Are the conclusions supported by the results?

(x) Can be improved

Please, read below comments to specific points.

Point 1: Lines 40-44. These sentences should be merged for improved reading flow. Suggested rewrite. “Soccer scouts often observe and select young players based on match performance [2], which can be affected by opponent rank, current match status, and other situational factors [3]. The use of standardized measurements (e.g., anthropometric, fitness, and technical skills) can assists coaches and practitioners in objectively examining soccer players' characteristics for subsequent selection [4-6].”

Response 1: We thank expert reviewer for his/her suggestion. Sentences were accordingly rewritten as follows:

“Soccer scouts often observe and select young players based on match performance [2], which can be affected by opponent rank, current match status, and other situational factors [3]. The use of standardized measurements (e.g., anthropometric, fitness, and technical skills) can assist coaches and practitioners in objectively examining soccer players' characteristics for subsequent selection [4-6].”

Point 2: Lines 52-55. Very confusing sentence in its current structure. Rewrite and separate into two separate sentences so the point(s) being made are clear. “Beyond the importance of muscle power and cardiorespiratory fitness, which are associated with physical performance (e.g., amount of high-speed distance) during competitive youth soccer games [3], limited attention has been paid to repeated sprint ability (RSA), which has been defined as the ability to recover and reproduce performance over subsequent sprints.” Additionally, does “amount of high-speed distance” refer to “total game running volume at    ?    velocity”?

Response 2: Sentence was split, re-phrased, and clarified as follows:

“Muscle power and cardiorespiratory fitness, which are associated with physical performance (e.g., amount of high-speed distance, >18 km·h-1 [14]) during competitive youth soccer games [3], were considered relevant. Beyond them, limited attention has been paid to repeated sprint ability (RSA), which has been defined as the ability to recover and reproduce performance over subsequent sprints.”

Point 3: Line 59. Add play after the word “match”.

Response 3: “play” was added after the word “match”. Now, sentence says:

“Importance of RSA is underlined by the fact that occurrence of repeated sprints is highly taxed during youth soccer with meaningful activity declines observed toward end of match play [15].”

Point 4: Line 62. Add an example for CR fitness (YO-YO intermittent test?).

Response 4: Suggested example was added to sentence that now says:

“However, research describing fitness profiles of young soccer players has predominantly focused on power-based abilities (e.g., jump, sprint, and change of direction [COD] ability) and cardiorespiratory fitness (e.g., Yo-Yo Intermittent Recovery Test) but with limited attention paid to RSA [18-21].”

Point 5: Line 77. Edit to read this for improved reading flow. “… evaluated at the start of the competitive (September 2017) for their…”

Response 5: Sentence was edited to read for improved reading flow as follows:

“All players were evaluated at start of competitive period (September 2017) for their respective clubs.”

Point 6: Line 80. Edit by adding the “…during the first…”

Response 6: Sentence was edited as follows:

“Anthropometric dimensions, dribbling skills, sprint time, COD, and RSA were assessed during the first visit in listed order.”

Point 7: Line 94. How is the person “trained”? Suggest replacing with “experienced” unless the trained is going to be explained such as trained by a specific organization.

Response 7: Sentence was re-worded as follows:

“The same experienced author measured body mass, height, sitting height, and four skinfolds (triceps, suprailiac, abdominal, and thigh).”

Point 8: Line 114. Edit sentence. “Split sprint times were measured at 5 and 15-m during a 20-m sprint test.”

Response 8: Sentence was re-worded as follows:

“Split sprint times measured at 5 and 15 m during a 20-m sprint test.”

Point 9: Line 118. Remove “Better” and just state, “The fastest sprints…”

Response 9: Sentence was re-worded as follows:

“The fastest sprints were used for further analysis.”

Point 10: Dribbling test would benefit from a figure that displays the planned path.

Response 10: A figure was added.

Point 11: Line 157. There should be a subsection title of Statistical Analysis.

Response 11: A new sub-heading entitled “Statistical Analysis” was added.

Point 12: Line 202. What does higher mean? Confusing sentence structure as it is difficult to figure out the point. Probably could be easily rewritten for clarity.

Response 12: I do apologise for the mistake: I meant “taller”. Sentence was re-worded as follows:

“The main findings of this study were that U17 selected soccer players were taller and presented, in general (except for CMJ), better functional capacities than their non-selected counterparts.”

Point 13: Line 205. Change of terms from “Selected” to “Elite”, be consistent with terms.

Response 13: Actually, for consistency purpose, I meant with “selected” (and “non-selected”) this study’s players. Since the very beginning of MS, since the title (“Anthropometric and functional profile of selected vs. non-selected young soccer players with special emphasis on repeated sprint ability”). Differently, I referred with “elite” (and “non-elite”) – or “top level”, etc. – other studies’ players. However, use of “selected” and “elite” terms was further checked and eventually amended throughout whole MS.

Point 14: Line 217. The 5-m and 15-m times represent acceleration not agility or quickness unless the athlete’s movement time to a stimulus was recorded.

Response 14: Sentence was re-worded as follows:

“On the other hand, U17 selected players were capable of higher acceleration over both 5 m and 15 m than non-selected players.”

Point 15: Lines 239-242. The sentence is confusing and needs to be rewritten for clarity and flow. “Nevertheless, previous research showed peak maximal oxygen uptake in elite league teams’ players higher than in first division probably due to elite league’s higher level of competition, which players were more likely exposed to, leading them thus to a more advanced development of aerobic capacities [34].”

Response 15: Sentence was split and re-worded as follows:

“Nevertheless, previous research showed that peak maximal oxygen uptake is higher in elite league teams’ players compared with first division. This is probably due to the fact that the level of competition of the elite league is higher. Its players are more likely exposed to such a higher level of competition and this leads them to a more advanced development of aerobic capacities [35].”

Point 16: Line 250-253. Rewrite these sentences for improved flow. “Main limitation of this study is lack of biological maturation assessment. In spite of being homogeneous in terms of chronological age, investigated 13-15 and 15-17 yrs old players were still in the process of growth and maturation and that could have interfered with their physical test measures.”

Response 16: We thank expert reviewer for his/her suggestion. Sentences were split and/or re-worded as follows:

“The main limitation of this study is the lack of an assessment of the biological maturation. This study investigated 13-15 and 15-17 yrs old players, who were homogeneous in terms of chronological age. Nevertheless, they were still in the process of growth and maturation and that could have interfered with their physical test measures.”

We hope that the manuscript has now reached the standard necessary for formal acceptance endorsement in Sports.

We look forward to hearing from you.

Best regards

Reviewer 2 Report

The purpose of the this study was to compare anthropometric and functional profile of 13-to-17 yrs old 21 soccer players according to their competitive level with special reference on repeated sprint ability. The study is interesting and may add value to the existing knowledge. Nevertheless, it has some flaws which have to be addressed. 

Abstract

line: 25. What do you mean by selected and non selected. Please specify. It would be better to distinguish your subjects as professionals and non-professionals.

Introduction

This is the greatest part of the article. Indeed, it is well-structured and straight to the point. Congratulations.

Methods

This part is also well-structured and clear. However i have a question related to the statistical analysis. Why the authors performed discriminant analysis?

Results and Discussion

The authors should be careful with some of their findings. This is due to the discriminant analysis. In my opinion, your main finding is that technical skills (dribbling skills)  distinguish selected (professionals)  and non-selected players. This can be obtained from the primary statistical analysis and it is of great importance. The rest of your results indicate that although you have some statistical differences (e.g. RSA mean time, 15-m sprint time), your effect size is relatively small and as such these variables/parameters likely have little effect on distinguishing players of different caliber, considering the age. I think that the abovementioned comments have to be discussed in the respective section and as such some of your findings should be modified (probably discriminant analysis is unnecessary). In other words, i think that your results are important, but the fact that you performed discriminant analysis while you have  such a low effect size values, may lead you in arbitrary conclusions.  Additionally, a  main problem here is related to maturation, which you did not measure ( although you report it in the limitations section). Moreover, i did not read if you performed any correlations. It would be interesting to see if dribbling skills are associated with some of your physical skills, and if these associations are similar to  professionals and non-professionals. 

Author Response

Is the research design appropriate?

(x) Can be improved

Please, read below comments to specific points.

Are the results clearly presented?

(x) Can be improved

Please, read below comments to specific points.

Are the conclusions supported by the results?

(x) Can be improved

Please, read below comments to specific points.

Point 1: line: 25. What do you mean by selected and non selected. Please specify. It would be better to distinguish your subjects as professionals and non-professionals.

Response 1: We thank expert reviewer for his/her suggestion. At Italian youth level, most of players cannot claim at all any professional-like status, i.e., it is very hard to foresee whether soccer will become their adulthood job or not. Therefore, I preferred using since the very beginning of MS, since the title (“Anthropometric and functional profile of selected vs. non-selected young soccer players with special emphasis on repeated sprint ability”), the selected (i.e., competing at national level) vs. non-selected ((i.e., competing at regional level) categorisation. However, this was further explained in Abstract as follows:

“Players were divided into selected (i.e., competing at national level, n=17 U15 and 47 U17) and non-selected (i.e., competing at regional level, n=43 U15 and 8 U17) group.”

Point 2: However, I have a question related to the statistical analysis. Why the authors performed discriminant analysis?

Response 2: Differences between groups might not be sufficiently explanative of why players are selected. Thus, the discriminant analysis was performed to identify the variables that better distinguish between competitive level and to evaluate the order of importance of these variables. In other words, it is a “filter” or a “further step” in addition to the comparative analysis.

Point 3: I did not read if you performed any correlations. It would be interesting to see if dribbling skills are associated with some of your physical skills, and if these associations are similar to professionals and non-professionals.

Response 3: We thank expert reviewer for his/her suggestion. Here, we kindly ask your understanding as this analysis is out of the scope of the article. Performing a correlational analysis among the wide set of variables we present in this study would result in an extremely broad and confused analysis. In other words, correlations between the variables we present might be appropriate for a separate study.

We hope that the manuscript has now reached the standard necessary for formal acceptance endorsement in Sports.

We look forward to hearing from you.

Best regards